

# Genome-wide identification of ABA receptor PYL family and expression analysis of *PYLs* in response to ABA and osmotic stress in *Gossypium*

Gaofeng Zhang[1,*], Tingting Lu[1,2,*], Wenwen Miao[1], Lirong Sun[1], Mi Tian[1], Ji Wang[1] and Fushun Hao[1]

[1] State Key Laboratory of Cotton Biology, Henan Key Laboratory of Plant Stress Biology, College of Life Sciences, Henan University, Kaifeng, Henan, China
[2] Henan University of Animal Husbandry and Economy, Zhengzhou, Henan, China
[*] These authors contributed equally to this work.

## ABSTRACT

Abscisic acid (ABA) receptor pyrabactin resistance1/PYR1-like/regulatory components of ABA receptor (PYR1/PYL/RCAR) (named PYLs for simplicity) are core regulators of ABA signaling, and have been well studied in *Arabidopsis* and rice. However, knowledge is limited about the PYL family regarding genome organization, gene structure, phylogenesis, gene expression and protein interaction with downstream targets in *Gossypium*. A comprehensive analysis of the *Gossypium* PYL family was carried out, and 21, 20, 40 and 39 PYL genes were identified in the genomes from the diploid progenitor *G. arboretum*, *G. raimondii* and the tetraploid *G. hirsutum* and *G. barbadense*, respectively. Characterization of the physical properties, chromosomal locations, structures and phylogeny of these family members revealed that *Gossypium* PYLs were quite conservative among the surveyed cotton species. Segmental duplication might be the main force promoting the expansion of *PYLs*, and the majority of the *PYLs* underwent evolution under purifying selection in *Gossypium*. Additionally, the expression profiles of GhPYL genes were specific in tissues. Transcriptions of many GhPYL genes were inhibited by ABA treatments and induced by osmotic stress. A number of GhPYLs can interact with GhABI1A or GhABID in the presence and/or absence of ABA by the yeast-two hybrid method in cotton.

## INTRODUCTION

Abscisic acid (ABA) is one of the most important phytohormones. It regulates multiple cellular processes including seed maturation and dormancy, seedling growth, leaf senescence and stomatal movement in plants (*Cutler et al., 2010*). Moreover, ABA play crucial roles in plant responses to various stresses such as drought, salinity, osmotic stress, extreme temperature, pathogen attack, and so on (*Cutler et al., 2010*; *Lee & Luan, 2012*). When plants are exposed to stresses, particularly dehydrate stress, the level of ABA in tissues prominently increases. It has been perceived that ABA can bind to the

Corresponding author
Fushun Hao, haofsh@126.com, haofsh@henu.edu.gn

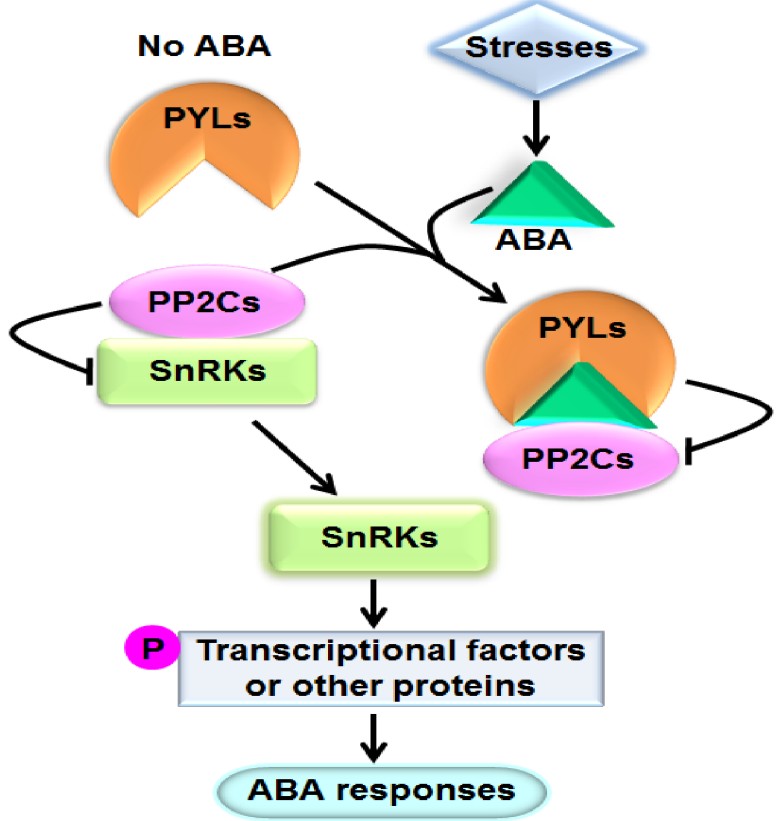

**Figure 1 A model showing the ABA key signal transduction pathway.** Arrows indicate positive regula-tion, bars indicate negative regulation, and P means phosphorylation.

ABA receptor pyrabactin resistance1/PYR1-like/regulatory components of ABA receptor (PYR1/PYL/RCAR) proteins (hereafter referred to as PYLs for simplicity). The formed complex interacts with and suppresses the activities of group A type 2C protein phosphatases (PP2Cs), the key negative regulators of ABA signaling. Consequently, PP2Cs unbind and activate the sucrose nonfermenting 1-related protein kinases (SnRKs) subfamily 2 (SnRK2s), the pivotal positive regulators of ABA signal cascade. SnRK2s subsequently phosphorylate and modulate downstream transcriptional factors or other proteins to cause physiological responses to ABA (*Fujii et al., 2009*) (Fig. 1). PYLs, PP2Cs and SnRK2s have been proved to be core components of ABA signaling. PP2Cs mainly include ABI1 (ABA insensitive 1), ABI2, HAB1 (Hypersensitive to ABA 1), HAB2, and so forth; and SnRK2s mainly consist of SnRK2.2, SnRK2.3 and SnRK2.6 in *Arabidopsis* (*Ma et al., 2009*; *Park et al., 2009*; *Lim et al., 2012*).

PYLs belong to the START (Star-related lipid-transfer) superfamily of ligand-binding proteins (*Ma et al., 2009*; *Park et al., 2009*). In *Arabidopsis*, 14 AtPYLs including AtPYR1 (*Arabidopsis thaliana* pyrabactin resistance1) and 13 AtPYLs (*Arabidopsis thaliana* PYR1-like1-13) were found (*Ma et al., 2009*; *Park et al., 2009*). Of which, AtPYR1, AtPYL1, AtPYL2 and AtPYL3 inhibit PP2Cs in a ABA dependent manner while AtPYL4, AtPYL5,

AtPYL6, AtPYL8, AtPYL9, AtPYL10 and AtPYL13 show suppression of PP2Cs in the absence of ABA (*Hao et al., 2011*; *Li et al., 2013*; *Miyakawa et al., 2013*). Genetic evidence indicated that AtPYR1, AtPYL1, AtPYL2, AtPYL4, AtPYL5 and AtPYL8 redundantly play roles in ABA-mediated seed germination, root growth and stomatal closure (*Park et al., 2009*; *Gonzalez-Guzman et al., 2012*). AtPYL5 and AtPYL9 regulate drought resistance (*Santiago et al., 2009*; *Zhao et al., 2016*). AtPYL9 also modulates leaf senescence (*Zhao et al., 2016*). Moreover, AtPYL8 and AtPYL9 were reported to function in root development (*Antoni et al., 2013*; *Zhao et al., 2014*; *Xing et al., 2016*). AtPYL6 and AtPYL13 were described to positively affect ABA inhibition of seed germination (*Fuchs et al., 2014*). In addition, it has been addressed that the orthologs of AtPYLs in rice, soybean, tomato, maize, wheat, poplar and *Brachypodium distachyon* play roles in growth and in response to stresses (*Kim et al., 2012*; *Bai et al., 2013*; *González-Guzmán et al., 2014*; *Tian et al., 2015*; *Fan et al., 2016*; *Gordon et al., 2016*; *Yu et al., 2016*; *Zhang et al., 2017*). Very recently, *Chen et al. (2017)* reported that overexpressing cotton *GhPYL10/12/26* in *Arabidopsis* increased ABA sensitivity during seed germination and seedling growth. *Liang et al. (2017)* also found that cotton *GhPYL9-11A* positively regulate drought tolerance in transgenic *Arabidopsis* plants. In recent years, many PYL gene families have been characterized at genome-wide levels in rice, grape, soybean and other plants (*Boneh et al., 2012*; *Kim et al., 2012*; *Bai et al., 2013*; *González-Guzmán et al., 2014*; *Tian et al., 2015*; *Fan et al., 2016*; *Gordon et al., 2016*; *Yu et al., 2016*; *Guo et al., 2017*; *Zhang et al., 2017*). Furthermore, the structure properties of AtPYLs in *Arabidopsis* have been intensively investigated. It is found that an ABA-binding pocket, three α-helix, seven β sheets and four loops are conserved in AtPYLs (*Ma et al., 2009*; *Park et al., 2009*; *Santiago et al., 2012*). Similar structures also exist in other plants. Furthermore, the expression patterns of PYL genes have been studied in tissues and in responding to exogenous ABA and diverse stresses in multiple plants (*Saavedra et al., 2010*; *Boneh et al., 2012*; *Kim et al., 2012*; *Bai et al., 2013*; *González-Guzmán et al., 2014*; *Tian et al., 2015*; *Fan et al., 2016*; *Gordon et al., 2016*; *Yu et al., 2016*; *Chen et al., 2017*; *Zhang et al., 2017*). These findings provide valuable information for researchers to further examine the functions of PYLs in ABA signaling in plants. However, knowledge about PYLs in *Gossypium* is scarce.

Cotton is the most important fiber crop and a key cash crop in the world. It provides the spinnable lint for the textile industry. It has been demonstrated that growth and development of cotton plants are seriously affected by a variety of environmental stresses such as drought, salinity and cold (*Allen, 2010*). These adverse stresses lead to significant decreases in the yield and quality of cotton fiber worldwide. Accordingly, it is of great importance to enhance the stress tolerance of cotton plants. One of the key strategies may be achieved via genetically engineering of PYL genes (*Kim et al., 2014*; *Lee et al., 2015*; *Yu et al., 2016*; *Zhao et al., 2016*; *Chen et al., 2017*; *Liang et al., 2017*). It is therefore essential for us to elucidate the functions and regulatory mechanisms of *Gossypium* PYLs. Here, we performed genome-wide and comprehensive analyses of the PYL family in *G. arboretum* ($A_2$) and *G. raimondii* ($D_5$), and their tetraploid species *G. hirsutum* ($AD_1$) and *G. barbadense* ($AD_2$). The expression patterns of PYL genes were studied in various tissues and in response to ABA and osmotic stress in *G. hirsutum*. Moreover, the interactions

between individual GhPYLs and GhABI1A or GhABI1D were measured in *G. hirsutum* by the yeast-two hybrid method. These results may pave the way for further investigating the functions of cotton PYLs in the future.

## MATERIALS AND METHODS

### Analysis of the PYL family in cotton

The amino acid sequences of 14 AtPYLs were searched in the genome sequence databases of *G. arboretum* (BGI-CGB v2.0 assembly genome), *G. raimondii* (JGI assembly v2.0 data.), *G. hirsutum* (NAU-NBI v1.1 assembly genome) (http://www.cottongen.org) and *G. barbadense* (http://database.chgc.sh.cn/cotton/index.html), respectively. The BLAST program was used with default setting ($E$-value $< e^{-10}$). The protein sequences of PYLs were pairwisely aligned applying the ClustalW software with default parameters (*Larkin et al., 2007*). Genes with questionable PYL annotations (i.e., having a typical PYL domain but low $E$-value or low coverage of a domain) were manually reanalyzed.

The properties of PYL proteins were assessed by the online tools ExPaSy (http://web.expasy.org/protparam/). The subcellular localizations of *Gossypium* PYLs were analyzed on the basis of the information from the website (http://www.csbio.sjtu.edu.cn/bioinf/Cell-PLoc). The locations of cotton PYL genes in chromosomes were determined by the MapInspect software (http://www.mybiosoftware.com/mapinspect-compare-display-linkage-maps.html) and their structures were identified by the GSDS (http://gsds.cbi.pku.edu.cn).

The conserved domains in PYLs were confirmed in NCBI (https://www.ncbi.nlm.nih.gov/Structure/cdd/wrpsb.cgi). The motifs in *Gossypium* PYLs were analyzed by MEME (*Bailey et al., 2009*).

### Phylogenetic analysis of PYLs

The PYL-related databases were downloaded from the websites for *Theobroma cacao* (http://cocoagendb.cirad.fr), *Ricinus communis* (http://castorbean.jcvi.org), *Vitis vinifera* (http://www.genoscope.cns.fr/spip/Vitis-vinifera-e.html), *Brachypodium distachyon* (http://plants.ensembl.org/Brachypodium_distachyon/Info/Index), *Oryza sativa* (http://rapdb.dna.affrc.go.jp) and *Arabidopsis thaliana* (http://www.arabidopsis.org), and four cotton plants described above. Multiple sequence alignments for the protein sequences of PYLs were carried out and phylogenetic trees were constructed following the alignment results using the neighbor joining method (Neighbor-Joining, NJ) and 1,000 bootstrap trials with the ClustalW tool (*Larkin et al., 2007*) and the MEGA 5 (http://www.megasoftware.net).

### Synteny and Ka/Ks analysis

The values of nucleotide substitution parameter Ka (non-synonymous) and Ks (synonymous) were counted based on the PAML program (*Yang, 1997*). The homologous genes were searched by the MCScanx software (*Wang et al., 2012*), and gene collinearity results were obtained by the CIRCOS program (*Krzywinski et al., 2009*). The syntenic maps of the PYL genes from *G. arboretum, G. raimondii* and *G. hirsutum* were constructed using the circos-0.69 ± 3 software package with default parameters (http://www.circos.ca).

## Expression analysis of PYL genes in tissues and in response to ABA or osmotic stress

To monitor expression levels of *PYLs* in tissues, samples of roots, stems and leaves were obtained from TM-1 cotton plants grown in soil for 21 d. Flowers were picked in the morning at the first day of anthesis, and fibers at the secondary cell wall stage (about 23 d post anthesis) were sampled from the ovules. To determine *PYL* transcript abundances in responding to ABA or osmotic stress, cotton plants grew in liquid 1/2 MS medium (*Murashige & Skoog, 1962*) in a growth chamber (day/night temperature cycle of 28 °C/26 °C, 14 h light/10 h dark, and about 50% relative humidity) for 3 weeks. Then, the plant leaves were sprayed with 100 µM ABA or the roots were treated with 10% PEG6000 (PEG6000 was added in the liquid medium) for 3, 6, 12 and 24 h, respectively. The plants that were not sprayed with ABA or were not treated with PEG6000 were as controls. The roots for treatments and controls were collected, frozen in liquid nitrogen and stored at −70 °C. About 0.1 g of samples were ground to a fine powder with a mortar and pestle in liquid nitrogen. Total RNA was extracted from the powder using RNA Pure Plant Kit's protocol (TIANGEN Company). The cDNA was obtained by M-MLV reserve transcriptase synthesis system (Promega, Madison, WI, USA) following the instructions in the Promega kit (https://tools.thermofisher.com/content/sfs/manuals/superscriptIII_man.pdf).

Quantitative real-time RT-PCR (qRT-PCR) experiments were carried out applying the cDNA, SYBR Green Master mix, the specific primers of cotton *PYL* genes (Table S1, the primer sequences were submitted to NCBI), and an ABI 7500 real-time PCR system. Cotton *UBQ7* was used as the internal control. Experiments were repeated at least three times.

## Yeast two-hybrid assay

Full-length sequences of gene *GhPYLs* and *GhABI1* were cloned into pGBKT7 and pGADT7 vectors, respectively, using primers listed in Table S2 (the primer sequences were submitted to NCBI). The resultant constructs were co-transformed into yeast strain AH109 (MATa, trp1-901, leu2-3, 112, ura3-52, his3-200, gal4Δ, gal80Δ, LYS2::GAL1UAS-GAL1TATAHIS3, GAL2UAS-GAL2TATA-ADE2, URA3::MEL1 UASMEL1TATA-lacZ, MEL1) according to the method described in page 18–21 in Yeast Protocols Handbook (*Clontech Laboratories Inc., 2009*). The cotransformants were plated on non-selective SD/-Leu/-Trp (synthetic dropout medium without Leu and Trp) agar plates and selective SD/-Leu/-Trp/-His/-Ade agar plates in the absence or presence of ABA. A cotransformed yeast spot was collected and diluted in sterile ddH$_2$O. The optical density (OD) value for the first solution of yeast colony was set 1. Serial 1:10 dilutions were generated, and 2 µL of the dilution was dropped on an agar plate to obtain one spot. The interactions between GhPYLs and GhABI1 were observed after 4 d of incubation at 30 °C. The experiments were repeated at least three times.

## RESULTS

### Genome-wide analysis of *PYLs* in four cotton species

To investigate the PYL family in the cotton genomes, the 14 AtPYL gene coding sequences and amino acid sequences were applied as queries to search against the cotton genome databases. A total of 21, 20, 40 and 39 PYL genes were identified in the genomes of two progenitor diploid species *G. arboretum* and *G. raimondii*, and their derived tetraploid species *G. hirsutum* and *G. barbadense*, respectively. The *PYLs* in the two diploid species were named based on their orthologous similarity to the 14 *AtPYLs* according to the methods described by *Mohanta et al. (2015)* in other genes. Briefly, the first letter of the genus (upper case) and the first letter of the species (lower case) followed by PYL and an AtPYL number were used to name a *Gossypium* PYL. The number of a *Gossypium* PYL was the same to that of its orthologous AtPYL sharing the most similarity in protein sequences to the *Gossypium* PYL. When more than one *Gossypium* PYLs had the same ortholog in *Arabidopsis*, additional numbers followed by a hyphen were applied to distinguish among paralogs of the *Gossypium* PYLs. The small number after the hyphen means high similarity of a cotton PYL to its corresponding AtPYL. The PYLs of the tetraploid cotton plants were denominated based on their phylogenetic relationship with those in *G. arboretum* and *G. raimondii*; and the last letter A or D meaned that the *PYL* was derived from A or D genome (Table S3). Therefore, the *Gossypium PYLs* in the four species were named *GaPYLs, GrPYLs, GhPYLs* and *GbPYLs*, respectively. We noticed that the *Gossypium* PYLs named following *AtPYL2, AtPYL4, AtPYL6, AtPYL9, AtPYL11, AtPYL12*, and *AtPYR1*. Moreover, most of these *AtPYLs* possessed not only one ortholog in *Gossypium*. For example, 8, 7, 14 and 13 homologs of *AtPYL9* were identified in genomes of *G. arboretum*, *G. raimondii*, *G. hirsutum* and *G. barbadense*, respectively (Table S3).

Analysis of the physical properties of the *Gossypium* PYL members revealed that these PYLs were highly conserved. Most PYLs had similar amino acid lengths, molecular weights (MWs), and theoretical isoelectric points (pI). Majority of PYLs in *Gossypium* possessed 177–222 amino acids. The MWs of the PYLs varied from 15.32 kDa to 32.22 kDa. The pIs of PYLs ranged from 4.73 to 9.51 with an average of 6.20. Most PYL proteins were predicted to locate in cytoplasm and/or nucleus (Table S3).

### Phylogenetic and structural analysis of cotton PYLs

To explore the evolutionary relationship of the PYLs among *G. arboreum*, *G. raimondii*, *G. hirsutum* and *G. barbadense*, an unrooted phylogenetic tree for the 120 PYLs was generated (Fig. 2A, Files S1–S2.). The PYLs can be divided into three subfamilies (I–III) based on the bootstrap values (>1,000) in the Neighbor-Joining (NJ) tree. Subfamily I had 41 members, which were orthologs of AtPYR1 (16) and AtPYL2 (25). Subfamily II consisted of 37 PYL genes. They were homologs of AtPYL4 (17), AtPYL6 (14), AtPYL11 (3) and AtPYL12 (3). Other 42 PYL members (homologues of AtPYL9) belonged to subfamily III (Fig. 2A). We found that the *Gossypium* homologs of the same AtPYL frequently clustered closely, indicating their more closed relative relationship. Besides, several PYLs including GaPYL2-2, GrPYL2-2, GhPYL2-2A, GrPYL2-2D, GbPYL2-2A, GbPYL2-2D and GbPYL2-2D′ appeared to be distant clades from other PYLs in *Gossypium* (Fig. 2A). This
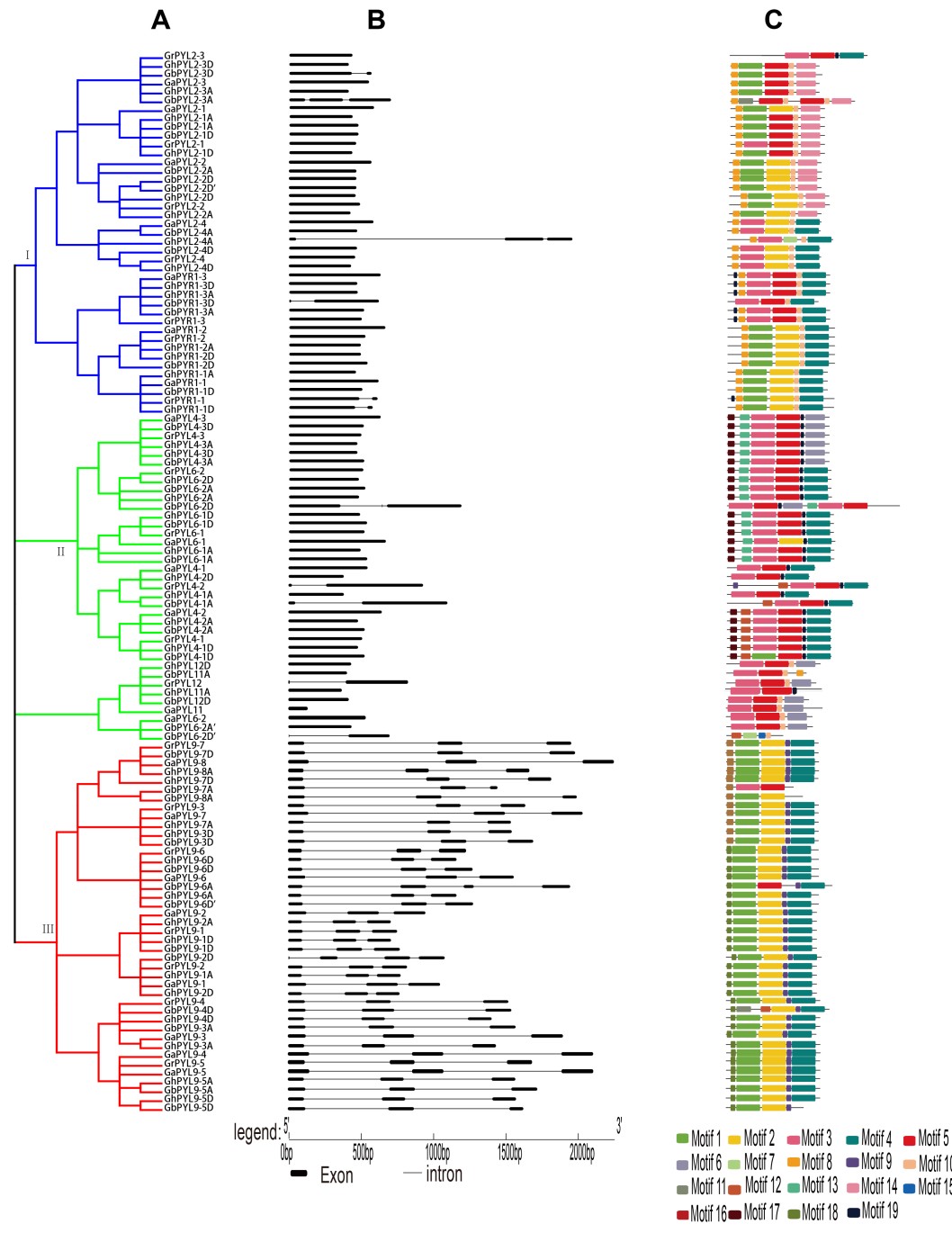

**Figure 2** **Phylogenetic relationships, gene architectures and conserved motifs of PYL genes in** ***Gossypium.*** (A) The phylogenetic tree was constructed by the Neighbor-Joining (NJ) method, with 1,000 bootstrap replicates. The dark blue, green and red lines show the subfamily I, II and III, respectively. (B) Exon/intron architectures of *Gossypium* PYL genes. The color boxes indicate exons, and the color lines represent introns. The sizes of exons and introns can be calculated following the scale at the bottom. (C) Distributions of conserved motifs. The motifs are indicated by 19 different color boxes.

suggests that these PYL2-2 members had relatively distant phylogenetic relationship with other *Gossypium* PYLs.

The exon/intron structures of the *Gossypium* PYL genes were studied. The results showed that the number of exons in the *PYLs* was 1–3. There was no intron in 66 *PYL* genes. Most of the remaining genes had two introns except that nine members had one intron (Fig. 2B). In addition, most *Gossypium* PYL genes clustered in the same subfamily had the similar number of exons and lengths of introns. For instance, vast majority of genes in subfamily I and II had only one exon, and 41 out of 42 genes in subfamily III had 3 exons and relatively long intron sequences (Fig. 2B). These results imply that the exon/intron organizations of cotton *PYLs* are closely related to the phylogenetic relationship of the genes.

To further clarify the diversity of motif compositions, the putative motifs in the 120 PYL proteins in subfamily I–III were analyzed by the MEME software. A total of 19 motifs designated as motif 1 to motif 19 were detected (Fig. 2C). The numbers of motifs in a PYL of subfamily I to III were 4–7, 4–7 and 3–6, respectively. Of the 19 motifs, motif 8 and 10 emerged in majority of PYLs in subfamily I, motif 3 and 5 were conservative among most PYLs in subfamily II; and motif 1, 2, 4 and 9 existed in overwhelming majority of subfamily III members. Most PYLs within the same subfamilies have very similar motif compositions and distributions, implying that PYLs within the same subfamilies probably share similar functions. Intriguingly, some motifs specifically existed in a particular subfamily or some PYLs. For example, motif 16 and 17 were only present in subfamily II, and motif 12 and 18 only belonged to members of subfamily III. Motif 11 was only seen in GbPYL2-3A and GbPYL9-4D. Motif 7 was only found in GhPYL2-4A and GbPYL6-2D′; and motif 15 was only observed in GbPYL6-2D′. These findings suggest that these motifs might play specific roles in the PYLs of that subfamily or in those PYLs. The detailed mechanisms need to be experimentally examined in the future.

## Conserved domains and amino acids of cotton PYLs

To better understand the structural similarity of *Gossypium* PYLs, the amino acid sequences of PYLs from *G. arboreum*, *G. raimondii* and *G. hirsutum* as well as those from the two diploid species and *G. barbadense* were aligned, respectively (Figs. S1 and S2). The results revealed that these PYLs had high sequence similarities. All the PYLs shared a similar helix-grip structure formed by seven β-sheets and three α-helices, and four identical conserved loops among the β-sheets and the α-helices (Figs. S1 and S2). Furthermore, many conserved amino acids (highlighted by red colour) were observed in the putative β-sheets, $\alpha$-helices as well as loops (Figs. S1 and S2), consistent with the structure of AtPYLs in *Arabidopsis* (Ma et al., 2009; Park et al., 2009). These conserved secondary structures and amino acids have been demonstrated to be essential for the functions of ABA receptors in *Arabidopsis*. For instance, the loops of CL1, CL2 and CL3 are essential for ABA bindings and the PYL-PP2C interactions (Zhang et al., 2015). Noteworthily, the conserved leucine (L) was replaced by methionine (M) in the CL2 loop, and the conserved arginine (R) was replaced by lysine (K) in the CL3 loop in GaPYL6-2, GbPYL6-2A′ and GbPYL6-2D′ (Figs. S1 and S2). Similarly, the conservative arginine (R) was replaced by methionine (M) in the CL3 loop in GhPYL9-5D (Fig. S1). These data hint that these PYLs may be

significantly different in ABA bindings and interactions with PP2C from other *Gossypium* PYLs.

## Chromosomal distributions of cotton *PYLs*

The localizations of the *Gossypium PYLs* in chromosomes were determined. Generally, PYL genes were unevenly distributed on multiple chromosomes (Fig. 3). The 21 *GaPYLs*, 20 *GrPYLs*, 40 *GhPYLs* and 39 *GbPYLs* separately placed on 10, 11, 21 and 18 chromosomes, respectively. Four genes were located on each of the Achr10 and Atchr05 chromosomes. Many chromosomes, each possessed three genes. These chromosomes included Achr06 from *G. arboretum*, Dchr07 and Dchr08 from *G. raimondii*, Atchr11, Atchr12, Dtchr05, Dtchr11 and Dtchr12 from *G. hirsutum*; and At′chr01, At′chr10, Dt′chr08, Dt′chr10, Dt′chr11 and Dt′chr12 from *G. barbadense* (Fig. 3). A large number of chromosomes individually owned two genes. These chromosomes were Achr03, Achr07, Achr09 and Achr13 in *G. arboretum*, Dchr04, Dchr06 and Dchr11 in *G. raimondii*, Atchr08, Atchr09, Atchr10, Dtchr08, Dtchr09 and Dtchr10 in *G. hirsutum*; and At′chr12, Dt′chr05 and Dt′chr09 in *G. barbadense*. Each of the rest chromosomes had one gene. The distributions of the PYL genes on individual chromosome were irregular. Some genes were located on the upper end of the chromosome arms, some placed on the lower end of the arms; whereas some lied in the region far from two end of the arms (Fig. 3). In addition, two genes in *G. arboretum* (*GaPYL2-1*, *GaPYL2-3*) and 6 genes in *G. barbadense* (*GbPYL4-2A*, *GbPYL9-3A*, *GbPYL9-5A*, *GbPYL9-6D′*, *GbPYL11A* and *GbPYL12D*) were present in scaffolds.

We compared the positions of the orthologs among *GaPYLs*, *GrPYLs* and *GhPYLs* in chromosomes. Unexpectedly, only a few *PYL* homologs localized in their corresponding homoeologous chromosomes. Similar results also happened among *GaPYLs*, *GrPYLs* and *GbPYLs* (Fig. 3). These findings hint that many complex conversion events of *PYL*-contained homoeologous chromosomes or of *PYLs* occurred among different *Gossypium* species during evolution.

## Synteny analysis of PYL genes

It has addressed that gene duplication events including tandem and segmental duplications play key roles in expanding gene family during the evolutionary process (*Cannon et al., 2004*). To gain insight into the genetic origins and evolution of the *Gossypium PYLs*, we analyzed the homologous gene pairs of *PYLs* among *G. arboreum*, *G. raimondii* and *G. hirsutum*. A sum of 88 collinearity blocks were identified between *G. Arboreum* and *G. raimondii*, and each block had one gene pair. Ninety-one collinearity blocks with 93 homologous pairs were detected between the At-genome and Dt-genome of *G. hirsutum*. Moreover, two homologous gene pairs between chromosome At11 and Dt11 (*GhPYL9-3A/GhPYL9-4D*, *GhPYR1-2A/GhPYR1-2D*), and chromosome At12 and Dt12 (*GhPYL9-5A/GhPYL9-5D*, *GhPYR1-3/GhPYR1-3D*) were found in two individual collinearity blocks. Additionally, 395 homologous gene pairs distributed in 385 collinearity blocks among *G. Arboreum*, *G. raimondii* and *G. hirsutum* (Fig. 4, Table S4). Of these blocks, one harbored three homologous gene pairs (*GrPYL9-1/GhPYL9-1D*, *GrPYL9-4/GhPYL9-4D*, *GrPYR1-2/GhPYR1-2D*). The block was between chromosome D07 in *G. raimondii* and

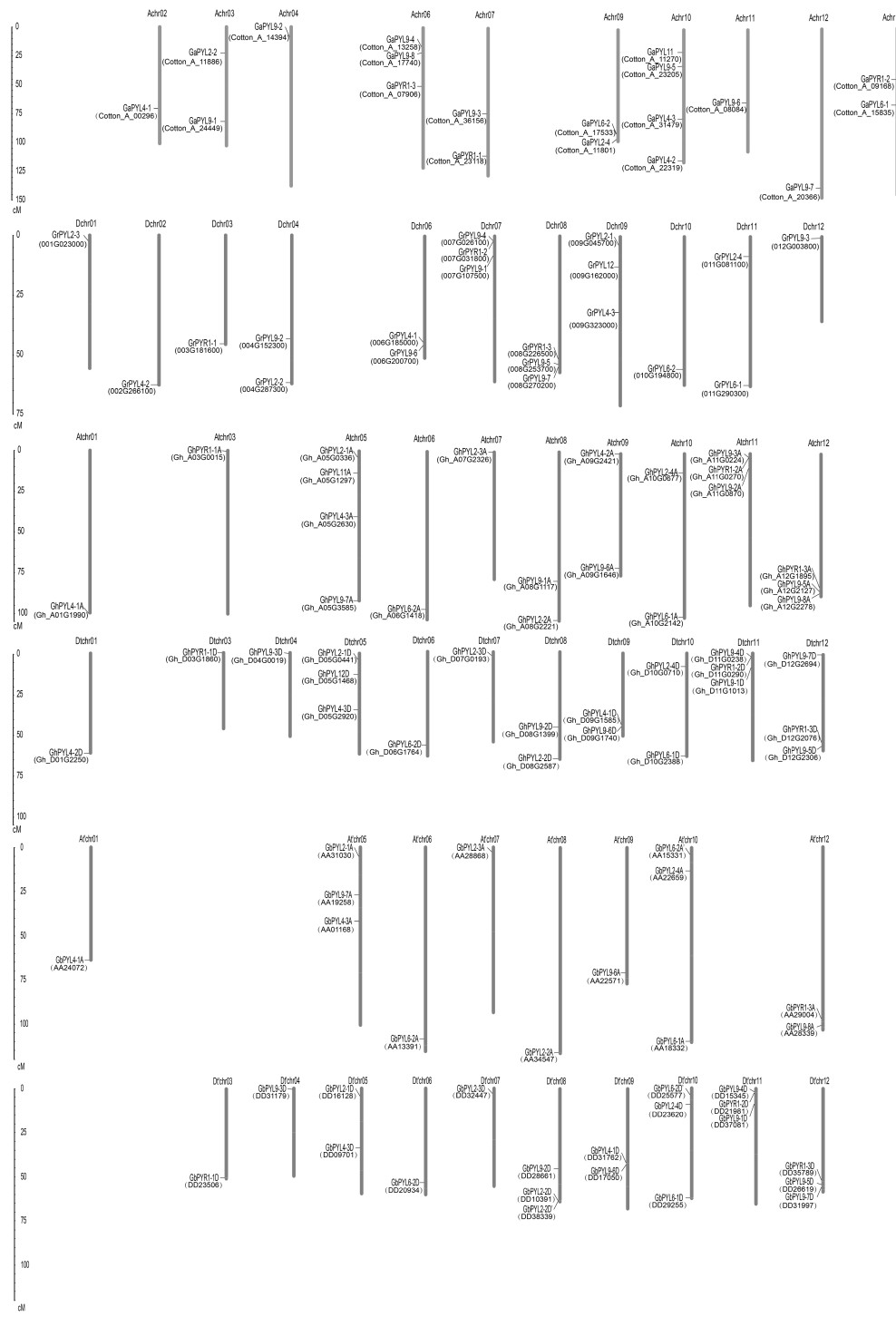

**Figure 3** **Distributions of *Gossypium* PYL genes on chromosomes.** *GaPYLs*, *GrPYLs*, *GhCBLs* and *GbPYLs* were from *G. arboreum*, *G. raimondii*, *G. hirsutum* and *G. barbadense*, respectively.

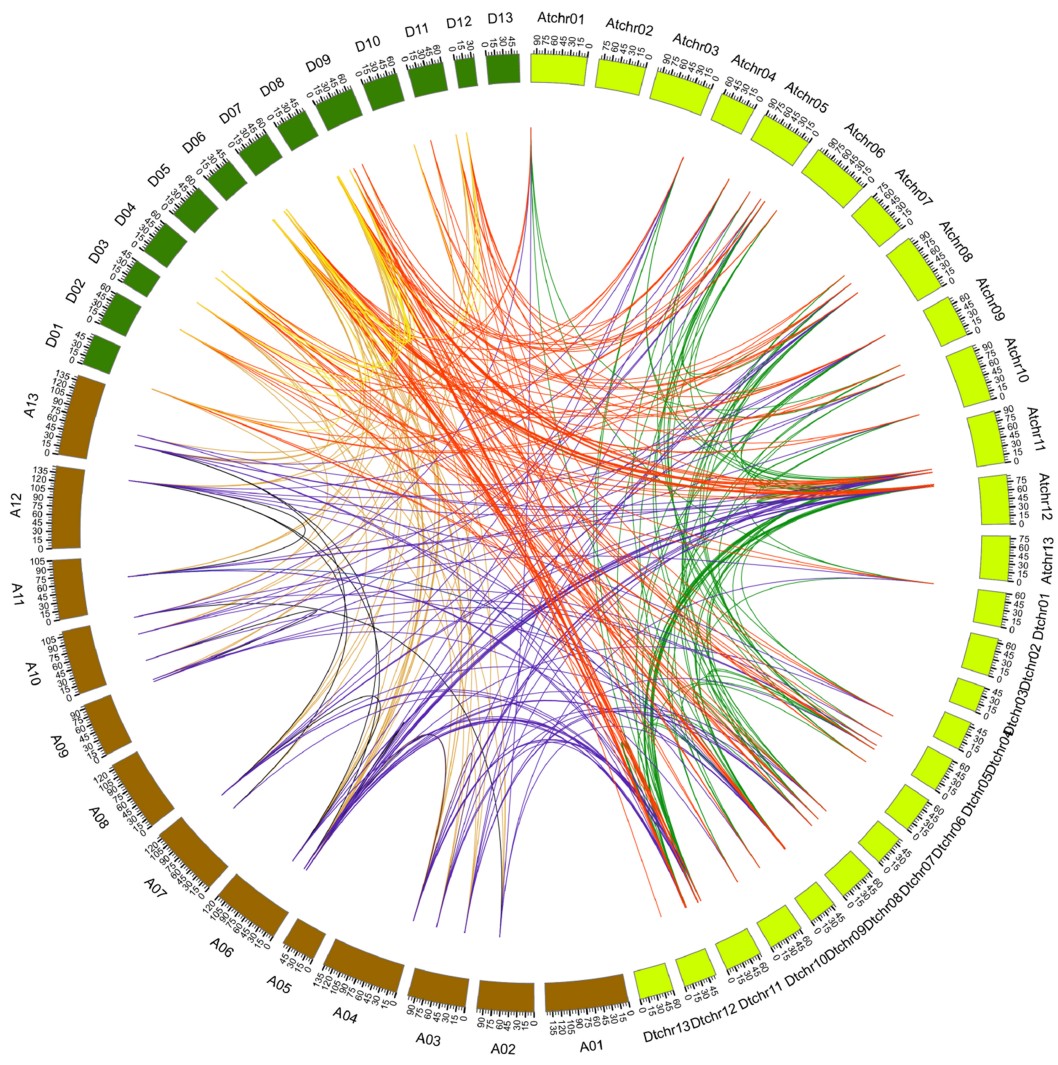

**Figure 4** **Genome-wide synteny analysis of PYL genes from *G. arboreum*, *G. raimondii* and *G. hirsutum*.** Synteny analysis between *G. hirsutum* and two diploid species *G. arboreum* and *G. raimondii.* Blue lines link gene pairs between *G. arboreum* and *G. hirsutum*, red lines connect gene pairs between *G. raimondii* and *G. hirsutum*, brown lines bridge gene pairs between *G. arboreum* and *G. raimondii,* black lines join gene pairs in *G. arboreum*, yellow lines link gene pairs in *G. raimondii,* and green lines connect gene pairs in *G. hirsutum*.

chromosome Dt11 in *G. hirsutum.* Other collinearity blocks individually possessed two homologous gene pairs (Table S4). No gene pair was implicated in tandem duplication. These results suggest that segmental duplications dominantly contribute to the generation of *Gossypium PYLs* during genetic evolution.

## Analysis of Ka/Ks values of *PYLs* in *G. arboreum*, *G. raimondii* and *G. hirsutum*

To further investigate the divergence and selection in duplication of *PYL* genes, the non-synonymous ($K$a), synonymous ($K$s) and $K$a/$K$s values were evaluated for the homologous

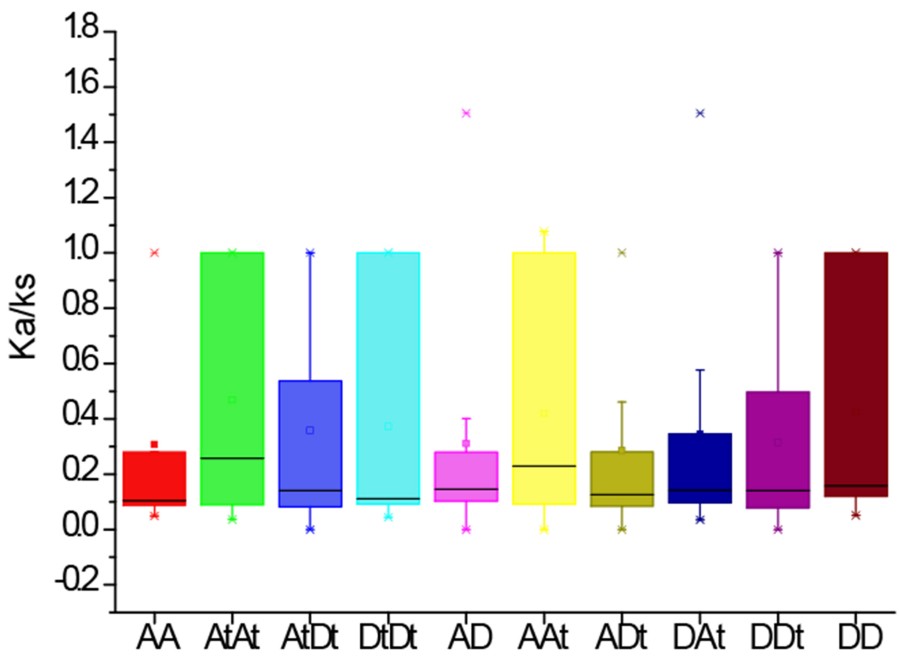

**Figure 5** The Ka/Ks values of the homologous PYL gene pairs among A genome of *G. arboretum* (A), D genome of *G. raimondii* (D), At and Dt subgenomes of *G. hirsutum* (At, Dt).

gene pairs among *G. arboreum*, *G. raimondii* and *G. hirsutum*. The results indicated that the average Ka/Ks values among homologous gene pairs of *PYLs* between genomes and/or subgenomes AA, AtAt, AtDt, DtDt, AD, AAt, ADt, DAt, DDt, DD were 0.31, 0.47, 0.36, 0.31, 0.42, 0.29, 0.34, 0.31 and 0.42, respectively (Fig. 5). The average Ka/Ks value of all the 301 gene pairs was 0.17, less than 1. These data suggest that these genes were mainly under the purifying selection during evolution. In contrast, the ratio of Ka to Ks between *GhPYL2-4A* and *GaPYL2-4* was 1.08, and that between *GhPYL9-8A* and *GrPYL9-7* was 1.51, indicating that the two gene pairs may generate under positive selection. Besides, the Ka/Ks values of many gene pairs were 1 (Table S5), implying that these genes are under neutral evolution.

## Phylogenetic relationship of PYLs in cotton and other plant species

To determine the evolutionary relationships of PYLs among the four cotton species and other plants, full-length amino acid sequences of the predicted PYLs were downloaded from the databases of *G. arboretum*, *G. raimondii*, *G. hirsutum*, *G. barbadense*, *T. cacao*, *R. communis*, *V. vinifera*, *B. distachyon*, *O. sativa* and *A. thaliana* (Table 1, Files S3–S5.), and a phylogenetic tree was constructed applying the neighbor-joining method and the MEGA 5.0 software. On the basis of the topologic structure, the PYL family members were classified into three subfamilies (I–III) (Fig. 6). Expectedly, vast majority of the *Gossypium* PYLs were closely clustered in a subfamily. Moreover, every subfamily contained PYLs from eudicots such as *Gossypium*, cocoa, castor and *Arabidopsis,* and from monocots like

Zhang et al. (2017), *PeerJ*, DOI 10.7717/peerj.4126

**Table 1** Gene nomenclature and homologs table.

| Homologs | G. arboretum | G. raimondii | G. hirsutum | G. barbadense | A. thaliana | T. cacao | R. communis | V. vinifera | B. distachyon | O. sativa |
|---|---|---|---|---|---|---|---|---|---|---|
| 1 | Cotton_A_13999 (GaPYL2-1) | 009G045700 (GrPYL2-1) | Gh_A05G0336 (GhPYL2-1A)/ Gh_D05G0441 (GhPYL2-1D) | AA31030 (GbPYL2-1A)/ DD16128 (GbPYL2-1D) | AT2G26040 (AtPYL2) | | 29794.m003335 (RcPYRL2) | GSVIVT01035362001 (VvPYRL2) | BRADI1G37810 (BdPYRL2)/ BRADI3G08580 (BdPYRL3) | Os06t0562200 (OsPYL2)/ Os02t0226801 (OsPYL3) |
| 2 | Cotton_A_11886 (GaPYL2-2) | 004G287300 (GrPYL2-2) | Gh_A08G2221 (GhPYL2-2A)/ Gh_D08G2587 (GhPYL2-2D) | AA34547 (GbPYL2-2A)/ DD10391 (GbPYL2-2D)/ DD38339 (GbPYL2-2D′) | AT2G26040 (AtPYL2) | | | | BRADI1G37810 (BdPYRL2)/ BRADI3G08580 (BdPYRL3) | Os06t0562200 (OsPYL2)/ Os02t0226801 (OsPYL3) |
| 3 | Cotton_A_05614 (GaPYL2-3) | 001G023000 (GrPYL2-3) | Gh_A07G2326 (GhPYL2-3A)/ Gh_D07G0193 (GhPYL2-3D) | AA28868 (GbPYL2-3A)/ DD32447 (GbPYL2-3D) | AT2G26040 (AtPYL2) | | | | BRADI1G37810 (BdPYRL2)/ BRADI3G08580 (BdPYRL3) | Os06t0562200 (OsPYL2)/ Os02t0226801 (OsPYL3) |
| 4 | Cotton_A_11801 (GaPYL2-4) | 011G081100 (GrPYL2-4) | Gh_A10G0677 (GhPYL2-4A)/ Gh_D10G0710 (GhPYL2-4D) | AA22659 (GbPYL2-4A)/ DD23620 (GbPYL2-4D) | AT2G26040 (AtPYL2)/ AT1G73000 (AtPYL3) | Thecc1EG029025 (TcPYRL3) | | | BRADI1G37810 (BdPYRL2)/ BRADI3G08580 (BdPYRL3) | Os06t0562200 (OsPYL2)/ Os02t0226801 (OsPYL3) |
| 5 | Cotton_A_00296 (GaPYL4-1) | 002G266100 (GrPYL4-2) | Gh_A01G1990 (GhPYL4-1A)/ Gh_D01G2250 (GhPYL4-2D) | AA24072 (GbPYL4-1A) | AT2G38310 (AtPYL4)/ AT5G05440 (AtPYL5)/ AT2G40330 (AtPYL6) | Thecc1EG021605 (TcPYRL4) | 29820.m001002 (RcPYRL3) | | BRADI2G22510 (BdPYRL4)/ BRADI1G16710 (BdPYRL5)/ BRADI1G65130 (BdPYRL6)/ BRADI2G53840 (BdPYLR7) | Os01t0827800 (OsPYL4)/ Os05t0473000 (OsPYL5)/ Os03t0297600 (OsPYL6) |
| 6 | Cotton_A_22319 (GaPYL4-2) | 006G185000 (GrPYL4-1) | Gh_A09G2421 (GhPYL4-2A)/ Gh_D09G1585 (GhPYL4-1D) | AA31755 (GbPYL4-2A)/ DD31762 (GbPYL4-1D) | AT2G38310 (AtPYL4)/ AT5G05440 (AtPYL5)/ AT2G40330 (AtPYL6) | Thecc1EG021605 (TcPYRL4) | 29820.m001002 (RcPYRL3) | | BRADI2G22510 (BdPYRL4)/ BRADI1G16710 (BdPYRL5)/ BRADI1G65130 (BdPYRL6)/ BRADI2G53840 (BdPYLR7) | Os01t0827800 (OsPYL4)/ Os05t0473000 (OsPYL5)/ Os03t0297600 (OsPYL6) |
| 7 | Cotton_A_31479 (GaPYL4-3) | 009G323000 (GrPYL4-3) | Gh_A05G2630 (GhPYL4-3A)/ Gh_D05G2920 (GhPYL4-3D) | AA01168 (GbPYL4-3A)/ DD09701 (GbPYL4-3D) | AT2G38310 (AtPYL4)/ AT5G05440 (AtPYL5)/ AT2G40330 (AtPYL6) | Thecc1EG042432 (TcPYRL5) | | | BRADI2G22510 (BdPYRL4)/ BRADI1G16710 (BdPYRL5)/ BRADI1G65130 (BdPYRL6)/ BRADI2G53840 (BdPYLR7) | Os01t0827800 (OsPYL4)/ Os05t0473000 (OsPYL5)/ Os03t0297600 (OsPYL6) |

Zhang et al. (2017), *PeerJ*, DOI 10.7717/peerj.4126

Table 1 (*continued*)

| Homologs | G. arboretum | G. raimondii | G. hirsutum | G. barbadense | A. thaliana | T. cacao | R. communis | V. vinifera | B. distachyon | O. sativa |
|---|---|---|---|---|---|---|---|---|---|---|
| 8 | Cotton_A_15835 (GaPYL6-1) | 011G290300 (GrPYL6-1) | Gh_A10G2142 (GhPYL6-1A)/ Gh_D10G2388 (GhPYL6-1D) | AA18332 (GbPYL6-1A)/ DD29255 (GbPYL6-1D) | AT2G38310 (AtPYL4)/ AT5G05440 (AtPYL5)/ AT2G40330 (AtPYL6) | | | | BRADI2G22510 (BdPYRL4)/ BRADI1G16710 (BdPYRL5)/ BRADI1G65130 (BdPYRL6)/ BRADI2G53840 (BdPYLR7) | Os01t0827800 (OsPYL4)/ Os05t0473000 (OsPYL5)/ Os03t0297600 (OsPYL6) |
| 9 | Cotton_A_17533 (GaPYL6-2) | | | AA15331 (GbPYL6-2A′)/ DD25577 (GbPYL6-2D′) | AT5G45860 (AtPYL11)/ AT5G45870 (AtPYL12)/ AT4G18620 (AtPYL13) | Thecc1EG029689 (TcPYL6) | | | | |
| 10 | Cotton_A_24449 (GaPYL9-1) | 004G152300 (GrPYL9-2) | Gh_A08G1117 (GhPYL9-1A)/ Gh_D08G1399 (GhPYL9-2D) | DD28661 (GbPYL9-2D) | AT4G01026 (AtPYL7)/ AT1G01360 (AtPYL9)/ AT4G27920 (AtPYL10) | Thecc1EG005169 (TcPYRL9) | | GSVIVT01027078001 (VvPYRL5) | BRADI2G32250 (BdPYRL8) | Os05t0213500 (OsPYRL11) |
| 11 | Cotton_A_14394 (GaPYL9-2) | 007G107500 (GrPYL9-1) | Gh_A11G0870 (GhPYL9-2A)/ Gh_D11G1013 (GhPYL9-1D) | DD37081 (GbPYL9-1D) | AT4G01026 (AtPYL7)/ AT1G01360 (AtPYL9)/ AT4G27920 (AtPYL10) | Thecc1EG005169 (TcPYRL9) | | GSVIVT01027078001 (VvPYRL5) | BRADI2G32250 (BdPYRL8)/ BRADI3G09580 (BdPYRL9) | Os02t0255500 (OsPYL10)/ Os05t0213500 (OsPYL11) |
| 12 | Cotton_A_36156 (GaPYL9-3) | 007G026100 (GrPYL9-4) | Gh_A11G0224 (GhPYL9-3A)/ Gh_D11G0238 (GhPYL9-4D) | AA40052 (GbPYL9-3A)/ DD15345 (GbPYL9-4D) | AT4G01026 (AtPYL7)/ AT1G01360 (AtPYL9)/ AT4G27920 (AtPYL10) | | 29742.m001442 (RcPYRL5) | GSVIVT01019517001 (VvPYRL3) | | |
| 13 | Cotton_A_13258 (GaPYL9-4)/ Cotton_A_23205 (GaPYL9-5) | 008G253700 (GrPYL9-5) | Gh_A12G2127 (GhPYL9-5A)/ Gh_D12G2306 (GhPYL9-5D) | AA38557 (GbPYL9-5A)/ DD26619 (GbPYL9-5D) | AT4G01026 (AtPYL7)/ AT1G01360 (AtPYL9)/ AT4G27920 (AtPYL10) | Thecc1EG016450 (TcPYRL7) | 29742.m001442 (RcPYRL5) | GSVIVT01019517001 (VvPYRL3) | BRADI3G09580 (BdPYRL9) | Os02t0255500 (OsPYL10) |
| 14 | Cotton_A_08084 (GaPYL9-6) | 006G200700 (GrPYL9-6) | Gh_A09G1646 (GhPYL9-6A)/ Gh_D09G1740 (GhPYL9-6D) | AA22571 (GbPYL9-6A)/ DD17050 (GbPYL9-6D)/ DD32170 (GbPYL9-6D′) | AT4G01026 (AtPYL7)/ AT1G01360 (AtPYL9)/ AT4G27920 (AtPYL10) | | 30169.m006525 (RcPYRL6) | | BRADI3G09580 (BdPYRL9) | Os02t0255500 (OsPYL10) |
| 15 | Cotton_A_20366 (GaPYL9-7) | 012G003800 (GrPYL9-3) | Gh_A05G3585 (GhPYL9-7A)/ Gh_D04G0019 (GhPYL9-3D) | DD31179 (GbPYL9-3D) | AT4G01026 (AtPYL7)/ AT1G01360 (AtPYL9)/ AT4G27920 (AtPYL10) | | 30169.m006525 (RcPYRL6) | | BRADI3G09580 (BdPYRL9) | Os02t0255500 (OsPYL10) |

Peerj

**Table 1** (*continued*)

| Homologs | G. arboretum | G. raimondii | G. hirsutum | G. barbadense | A. thaliana | T. cacao | R. communis | V. vinifera | B. distachyon | O. sativa |
|---|---|---|---|---|---|---|---|---|---|---|
| 16 | Cotton_A_17740 (GaPYL9-8) | 008G270200 (GrPYL9-7) | Gh_A12G2278 (GhPYL9-8A)/ Gh_D12G2694 (GhPYL9-7D) | DD31997 (GbPYL9-7D) | AT4G01026 (AtPYL7)/ AT1G01360 (AtPYL9)/ AT4G27920 (AtPYL10) | | 30190.m010824 (RcPYRL7) | | BRADI3G09580 (BdPYRL9) | Os02t0255500 (OsPYL10) |
| 17 | Cotton_A_11270 (GaPYL11) | 009G162000 (GrPYL12) | Gh_A05G1297 (GhPYL11A)/ Gh_D05G1468 (GhPYL12D) | AA37880 (GbPYL11A)/ DD17498 (GbPYL12D) | AT5G45860 (AtPYL11)/ AT5G45870 (AtPYL12)/ AT4G18620 (AtPYL13) | | | | | |
| 18 | Cotton_A_23118 (GaPYR1-1) | 003G181600 (GrPYR1-1) | Gh_A03G0015 (GhPYR1-1A)/ Gh_D03G1860 (GhPYR1-1D) | DD23506 (GbPYR1-1D) | AT4G17870 (AtPYR1)/ AT5G46790 (AtPYL1) | | | GSVIVT01013161001 (VvPYRL1) | BRADI3G34070 (BdPYRL1) | Os02t0255500 (OsPYL1) |
| 19 | Cotton_A_09168 (GaPYR1-2) | 007G031800 (GrPYR1-2) | Gh_A11G0270 (GhPYR1-2A)/ Gh_D11G0290 (GhPYR1-2D) | DD21981 (GbPYR1-2D) | AT4G17870 (AtPYR1)/ AT5G46790 (AtPYL1) | Thecc1EG015719 (TcPYRL1) | 29827.m002533 (RcPYRL1) | GSVIVT01013161001 (VvPYRL1) | BRADI3G34070 (BdPYRL1) | Os02t0255500 (OsPYL1) |
| 20 | Cotton_A_07906 (GaPYR1-3) | 008G226500 (GrPYR1-3) | Gh_A12G1895 (GhPYR1-3A)/ Gh_D12G2076 (GhPYR1-3D) | AA29004 (GbPYR1-3A)/ DD35789 (GbPYR1-3D) | AT4G17870 (AtPYR1)/ AT5G46790 (AtPYL1) | | | | BRADI3G34070 (BdPYRL1) | Os02t0255500 (OsPYL1) |
| 21 | | 010G194800 (GrPYL6-2) | Gh_A06G1418 (GhPYL6-2A)/ Gh_D06G1764 (GhPYL6-2D) | AA13391 (GbPYL6-2A)/ DD20934 (GbPYL6-2D) | AT2G38310 (AtPYL4)/ AT5G05440 (AtPYL5)/ AT2G40330 (AtPYL6) | | 29729.m002290 (RcPYRL4) | | BRADI1G16710 (BdPYRL5)/ BRADI1G65130 (BdPYRL6) | Os01t0827800 (OsPYL4)/ Os05t0473000 (OsPYL5)/ Os03t0297600 (OsPYL6) |
| 22 | | | | AA19258 (GbPYL9-7A)/ AA28339 (GbPYL9-8A) | AT4G01026 (AtPYL7)/ AT5G53160 (AtPYL8)/ AT1G01360 (AtPYL9)/ AT4G27920 (AtPYL10) | Thecc1EG015359 (TcPYRL8) | | GSVIVT01028704001 (VvPYRL4) | | |

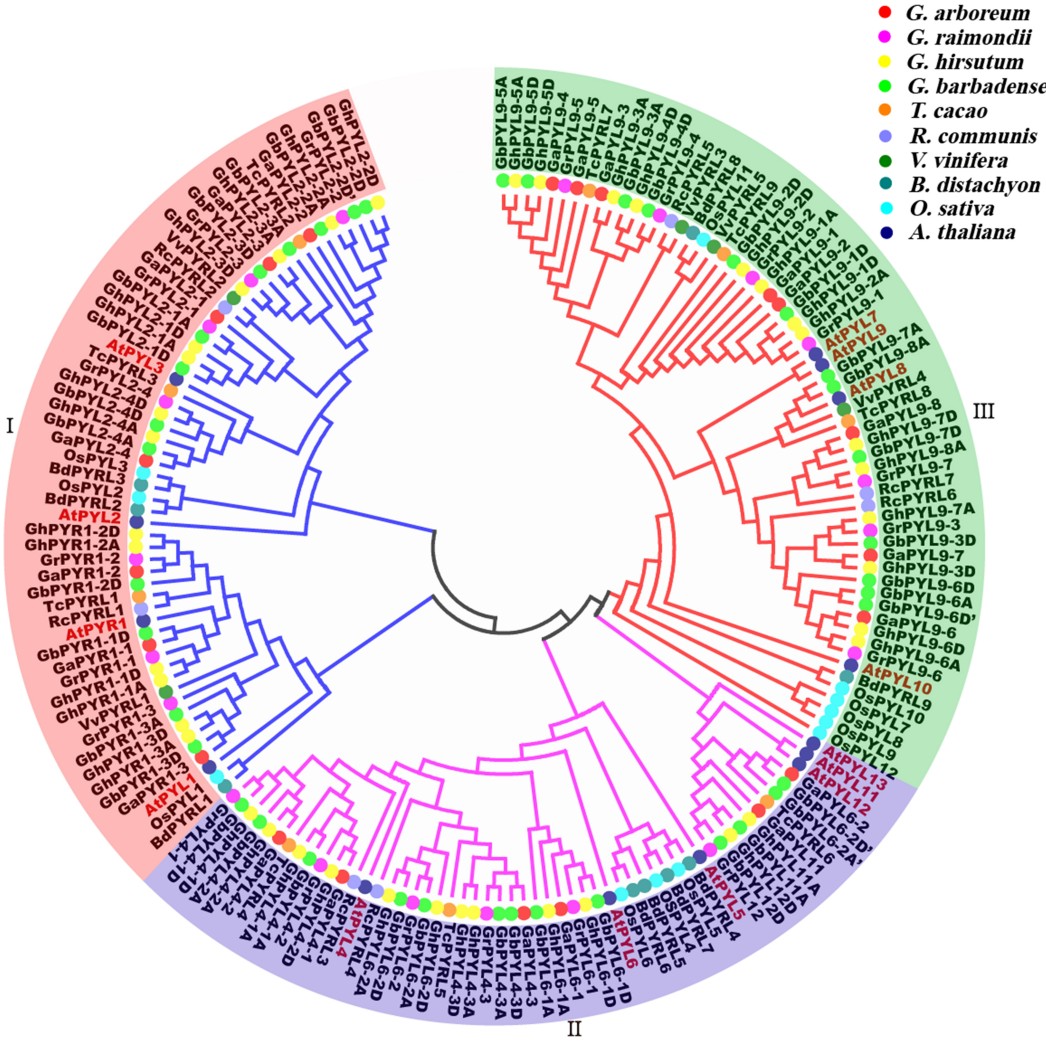

**Figure 6**  Phylogenetic tree of PYLs in cotton and other plant species.

*B. distachyon* and *O. sativa*, hinting that these PYLs generated before the divergence from eudicots and monocots. In addition, numerous PYLs from cotton species clustered more closely with those from cocoa than from other plants (Fig. 6), reflecting closer relationship of *Gossypium* PYLs with cacao PYLs relative to other plant PYLs.

## Expression of *GhPYL* genes in tissues

To further understand the roles of cotton PYLs in diverse organs, the expression patterns of all the 40 *GhPYL* genes were monitored by qRT-PCR. Of these *GhPYLs*, 22 genes were preferentially expressed in the flower, 10 genes were dominantly expressed in the root. Moreover, three genes including *GhPYR1-3D*, *GhPYL2-2A* and *GhPYL2-2D* were highly expressed in the fiber. The transcripts of *GhPYR1-3A*, *GhPYL4-3A*, *GhPYL4-3D*, *GhPYL9-1A* and *GhPYL12D* were relatively abundant in the stem (Fig. 7). These results indicate that some ABA receptors may specially function in a unique tissue. Besides, more

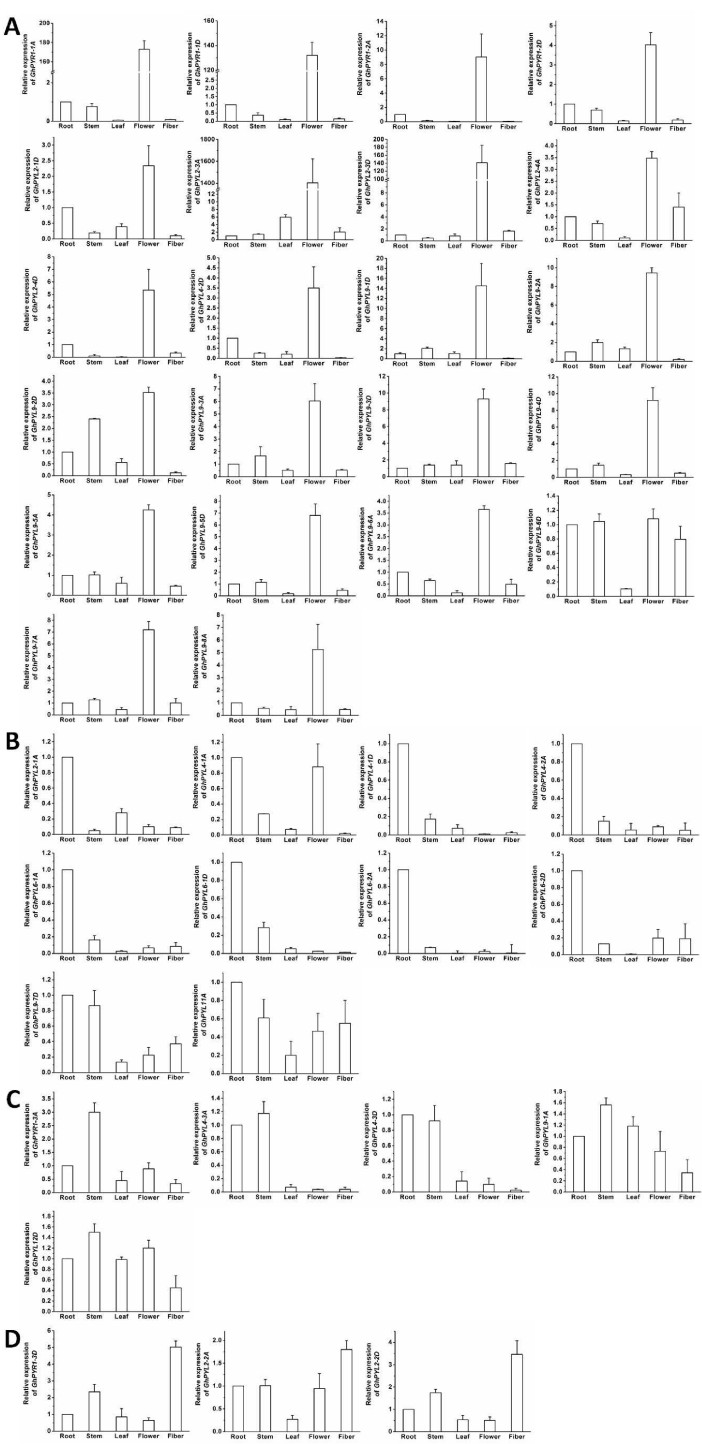

**Figure 7  Expression of *GhPYL* genes in tissues of cotton.** The genes preferentially expressed in flowers (A), roots (B), stems (C) and fibers (D) are shown. Gene *GhUBQ7* was used as the internal control. The expression value of a gene in roots was set as 1. The data are mean ± SE.

*GhPYLs* were predominantly expressed in the flower and the root, suggesting that ABA receptors may play important roles in fiber formation and in response to abiotic stresses in cotton.

## Expression profiles of *GhPYLs* in responses to ABA and osmotic stress

The expression patterns of the *GhPYL* genes in roots were monitored after treatments with 100 μM ABA or 10% PEG6000 for different period of time. In general, the transcriptional levels of most *GhPYLs* were decreased in responding to ABA (Fig. 8), and moderately enhanced in response to osmotic stress (Fig. 9). After treatment with ABA for 3 h or 6 h, expression of many genes for example *GhPYR1-1A*, *GhPYR1-3D*, *GhPYL9-5A* significantly reduced, then increased slowly at 12 h or 24 h. Expression of some genes like *GhPYL4-1D*, *GhPYL4-2A* and *GhPYL6-1D* continually decreased in response to ABA. However, the transcriptional abundances of some genes increased upon ABA treatment. These genes included *GhPYL2-2A*, *GhPYL2-3A*, *GhPYL2-3D*, *GhPYL4-3D*, *GhPYL6-2A*, *GhPY6-2D*, *GhPYL9-1D* and *GhPYL12D*. By contrast, the expression of *GhPYL9-2D* was not influenced by ABA treatment (Fig. 8). These results indicate that *GhPYLs* play differential roles in perceiving ABA signals.

The expression of most *GhPYLs* was upregulated by PEG treatments for 12 h and/or 24 h. By contrast, the transcriptional levels of *GhPYR1-3*, *GhPYR1-3D*, *GhPYL2-1A* and *GhPYL2-4D* were diminished whereas those of *GhPYR1-2A* and *GhPYR1-2D* were unchanged after exposure to high concentration of PEG6000 (Fig. 9). These results suggest that a great number of *GhPYLs* have different responses to ABA relative to osmotic stress in cotton.

## Many GhPYLs interact with GhABI1A or GhABI1D

PYLs have been addressed to transduce ABA signals to downstream targets through selectively interplaying with clade A PP2C proteins such as ABI1 and ABI2 in plants (*Cutler et al., 2010*; *Joshi-Saha, Valon & Leung, 2011*). We therefore examined the interactions between GhPYLs and GhABI1A (Gh_A07G0123) or GhABI1D (Gh_D07G2383) in the absence or presence of ABA by yeast-two hybrid method. Twenty-five out of fourty *GhPYLs* were cloned because of high similarity of CDS sequences among different GhPYL members. In the absence of ABA, 9 GhPYLs individully interplayed with GhABI1A, and 8 GhPYLs respectively interacted with GhABI1D (Table 2). Among them, GhPYR1-1D, GhPYL6-2D, GhPYL9-1A and GhPYL9-5D displayed relative weak interactive signals with GhABI1D. GhPYL6-2A and GhPYL9-4D showed strong interaction signals with GhABI1D (Fig. 10). When supplied with 10 μM ABA in the medium, 17 GhYPLs could interact with GhABI1A, and 14 GhPYLs could interplay with GhABI1D. We observed that many interactions were ABA-dependent, and numerous interactions were ABA-independent (Fig. 10, Table 2). Moreover, the interactive intensities between GhPYL6-2A and GhABI1A as well as between GhPYL9-5D and GhABI1D were increased by ABA. Interestingly, the interaction signal between GhPYL4-2A and GhABI1A was slightly weakened by ABA. These results suggest that GhPYLs differentially or specifically bind to GhABI1 in response to stresses, reflecting the diverse interacting modes among GhPYLs, GhABI1 and ABA in cotton.

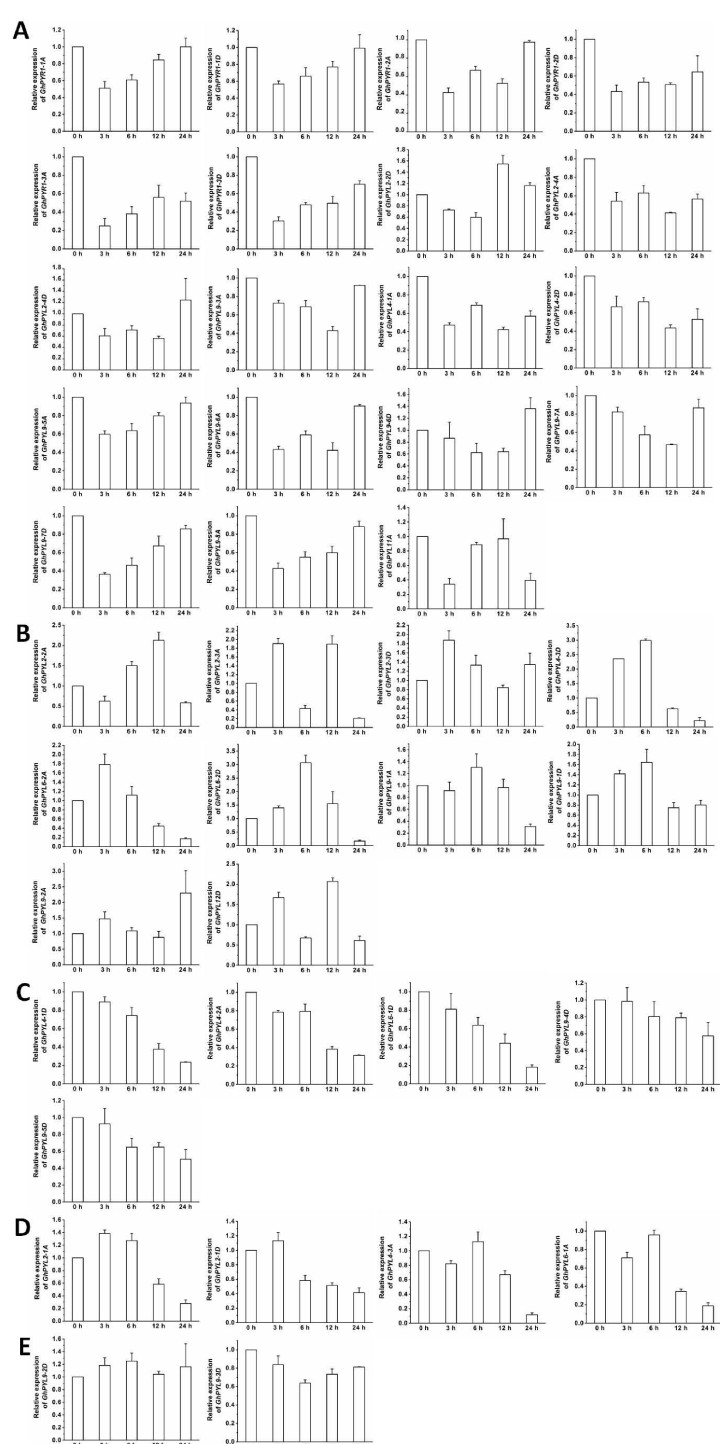

**Figure 8 Expression of *GhPYL* genes in response to ABA. The relative expression of *GhPYLs* was examined after treatments with 100 μM ABA for indicated period of time.** The expression levels of the genes were markedly decreased at 3 h or 6 h but increased at 12 h or 24 h (A), were continually increased (B), decreased (C), and increased at 3 h or 6 h but decreased at 12 h or 24 h (D), and were not altered (E). Cotton gene *GhUBQ7* was applied as the internal control. The gene expression value at 0 h was set as 1. The values are mean ± SE.

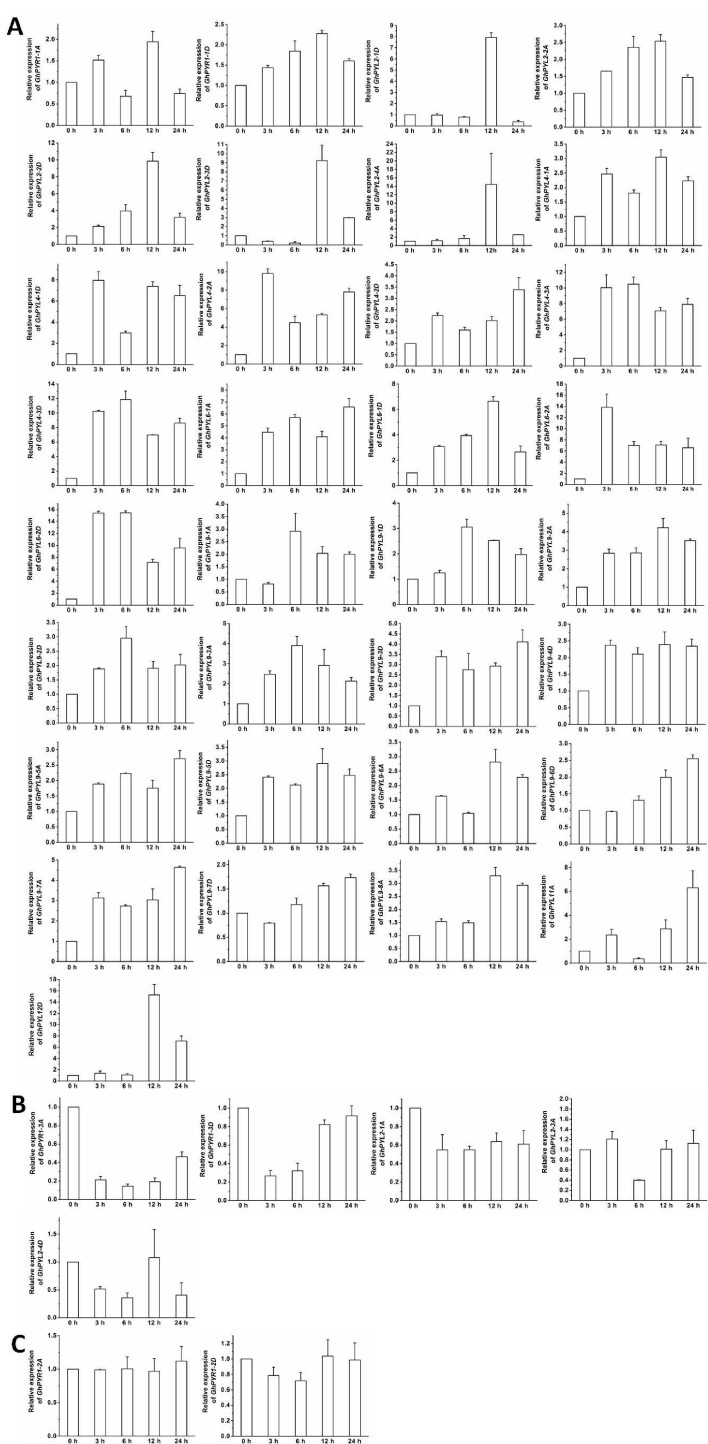

**Figure 9 Expression of *GhPYL* genes in response to osmotic stress.** The relative expression of *GhPYLs* was analyzed after treatments with 10% PEG6000 for indicated period of time. The transcriptional levels of the genes were remarkably increased (A), decreased (B), and not altered (C). Gene *GhUBQ7* was applied as the internal control. The gene expression value at 0 h was set as 1. The values are mean ± SE.

**Table 2** Schema of interactions between GhPYLs and GhABI1A/GhABI1D.

| Genes | Gene ID | −ABA | | +ABA | |
|---|---|---|---|---|---|
| | | GhABI1A (Gh_A07G0123) | GhABI1D (Gh_D07G2383) | GhABI1A (Gh_A07G0123) | GhABI1D (Gh_D07G2383) |
| GhPYR1-1A | Gh_A03G0015 | × | × | √ | × |
| GhPYR1-1D | Gh_D03G1860 | √ | √ | √ | √ |
| GhPYR1-2A | Gh_A11G0270 | × | × | × | × |
| GhPYR1-2D | Gh_D11G0290 | × | × | √ | √ |
| GhPYR1-3A | Gh_A12G1895 | √ | × | √ | × |
| GhPYL2-1A | Gh_A05G0336 | × | × | √ | √ |
| GhPYL2-2A | Gh_A08G2221 | × | × | √ | √ |
| GhPYL2-2D | Gh_D08G2587 | × | × | √ | √ |
| GhPYL2-3D | Gh_D07G0193 | × | × | × | × |
| GhPYL4-1A | Gh_A01G1990 | × | × | √ | √ |
| GhPYL4-2A | Gh_A09G2421 | √ | √ | √ | √ |
| GhPYL4-2D | Gh_D01G2250 | × | × | √ | √ |
| GhPYL4-3A | Gh_A05G2630 | × | × | × | × |
| GhPYL6-1D | Gh_D10G2388 | √ | × | √ | × |
| GhPYL6-2A | Gh_A06G1418 | √ | √ | √ | √ |
| GhPYL6-2D | Gh_D06G1764 | √ | √ | √ | √ |
| GhPYL9-1A | Gh_A08G1117 | × | √ | × | √ |
| GhPYL9-2A | Gh_A11G0870 | √ | × | √ | × |
| GhPYL9-3D | Gh_D04G0019 | × | × | × | × |
| GhPYL9-4D | Gh_D11G0238 | √ | √ | √ | √ |
| GhPYL9-5D | Gh_D12G2306 | × | √ | √ | √ |
| GhPYL9-6A | Gh_A09G1646 | √ | √ | √ | √ |
| GhPYL9-6D | Gh_D09G1740 | × | × | × | × |
| GhPYL9-7D | Gh_D12G2694 | × | × | × | × |
| GhPYL11A | Gh_A05G1297 | × | × | × | × |

**Notes.**
The symbols "√" and "×" respectively mean "interaction" and "no interaction" between two proteins. "−ABA" and "+ABA" represents experiments performed in the absence or presence of 10 μM ABA.

# DISCUSSION

ABA receptor PYLs are key regulators of ABA signaling, and have been investigated in many plants in recent years (*Boneh et al., 2012*; *Kim et al., 2012*; *Bai et al., 2013*; *González-Guzmán et al., 2014*; *Fan et al., 2016*; *Gordon et al., 2016*; *Zhang et al., 2015*; *Chen et al., 2017*; *Guo et al., 2017*; *Liang et al., 2017*). However, knowledge about phylogenesis and the roles of most PYLs in *Gossypium* is limited. In this report, we identified 21, 20, 40 and 39 PYL genes in genomes of *G. Arboreum*, *G. raimondii*, *G. hirsutum* and *G. barbadense*, respectively (Table S3). Compared with the number of PYLs in the reported plants including an *Arabidopsis* (14), a rice (13), a barley (9), a maize (11), a tomato (15), a *Brachypodium distachyon* (12), a soybean (23), a poplar (14) and a rubber tree (14), that in *Gossypium*, especially in the tetraploid species, was very large (*Kim et al., 2012*; *Bai et al., 2013*; *González-Guzmán et al., 2014*; *He et al., 2014*; *Seiler et*
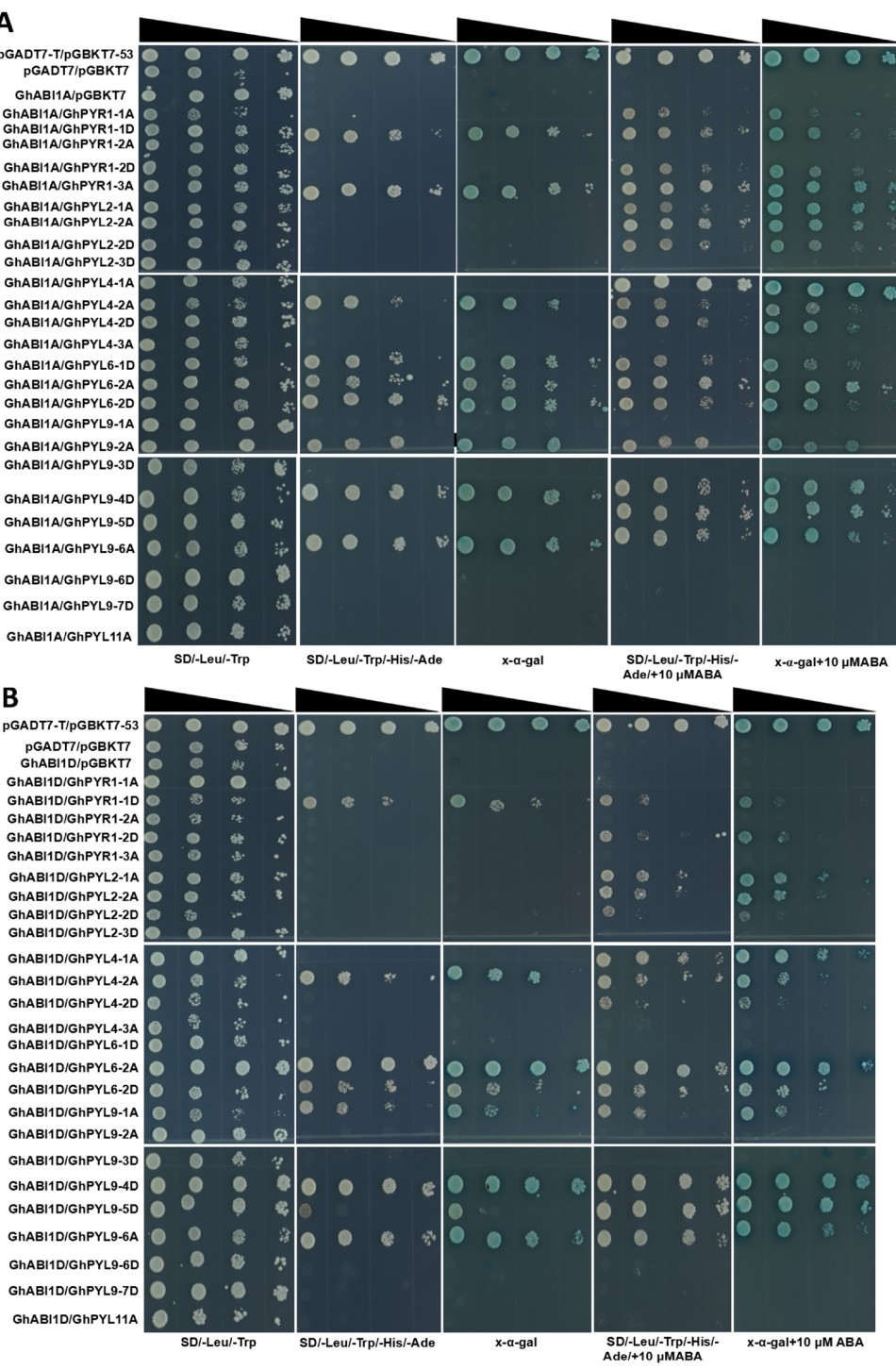

**Figure 10   Analysis of interactions between GhPYLs and GhABI1A or GhABI1D.** Interactions between GhPYLs and GhABI1A (A), and between GhPYLs and GhABI1D (B) were examined by yeast two-hybrid method. The reduced cell densities in the dilution series are revealed by narrowing triangles when proceeding from left to right.

*al., 2014*; *Fan et al., 2016*; *Yu et al., 2016*; *Zhang et al., 2017*; *Guo et al., 2017*), indicating more complex ABA responses likely happen in *Gossypium*. *Chen et al. (2017)* identified 27 GhPYLs (*Chen et al., 2017*), significantly less than 40 members as described in this study. The main reason may be that the genome sequence database of *G. hirsutum* they chose to search homologs of AtPYLs was different from what we did. *Chen et al. (2017)* used the database from the website http://cgp.genomics.org.cn/page/species/index.jsp whereas we applied that from https://www.cottongen.org. *Liang et al. (2017)* reported that there exist 22, 22, 44 and 36 PYLs in *G. arboretum*, *G. raimondii*, *G. hirsutum* and *G. barbadense*, respectively, inconsistent with our results. The possible reason might be the standard we used to determine PYLs in *Gossypium* was slightly different from what they did. For instance, we also surveyed 44 putative GhPYLs. However, three of them (Gh_A10G0340, Gh_D10G0346 and Gh_D08G1226) lack the PYR_PYL_RCAR_like domain of PYLs, and Gh_Sca112916G01 encodes a too small protein. Accordingly, the four proteins were eliminated in this report.

Of the 40 *GhPYLs* we identified, 20 and 20 were from A and D subgenomes, respectively; and every member had its corresponding ortholog in *a G. arboretum* and a *G. raimondii*. Likewise, most *GbPYLs* individually possessed their corresponding orthologs in a *G. arboretum* and a *G. raimondii*. However, both *GbPYL6-2A* and *GbPYL6-2D* had high sequence similarity to *GrPYL6-2*. Similar cases occurred in *GbPYL6-2A′* and *GbPYL6-2D′*, *GbPYL2-2D* and *GbPYL2-2D′* as well as in *GbPYL9-6D* and *GbPYL9-6D′*. The three pairs of genes seemed to derived from *GaPYL6-2*, *GrPYL2-2* and *GrPYL9-6*, respectively (Table S3). These findings suggest that gene replication events contribute to the generation of these *GbPYL* pairs in sea island cotton. Moreover, *GaPYL9-4* had no corresponding homologous gene in *GhPYLs*, and some *GaPYLs* (*GaPYR1-1*, *GaPYR1-2*, *GaPYL9-1*, *GaPYL9-2*, *GaPYL9-4*) and *GrPYL4-2* did not share corresponding homologous genes in *GbPYLs* (Table S3), implying that gene loss events occurred during the evolution processes of *Gossypium PYLs*.

We observed that some *AtPYLs* (for example *AtPYL9*) had several homologues of *GaPYLs*, *GrPYLs*, *GhPYLs* or *GbPYLs* (Table S3), hinting that selective gene replications likely play essential roles in the generation of the *Gossypium PYLs* during evolution.

The PYLs of *G. arboreum* and *G. raimondii* exhibited similar physical properties to those of *G. hirsutum* and *G. barbadense* (Table S3). This suggests that the functions of the PYLs in the four cotton plants were conserved during the processes of genetic evolution. The predicted subcellular localizations of PYLs in *Gossypium* were cytoplasm and/or nucleus, in line with the locations of PYLs from *Arabidopsis*, rice, and soybean plants (*Ma et al., 2009*; *Bai et al., 2013*; *Tian et al., 2015*). These data indicate that the functions of PYLs are conservative among different plant species.

We saw that each subfamily of *Gossypium PYLs* contained several members of *G. arboreum*, *G. raimondii*, *G. hirsutum* and *G. barbadense*. Moreover, the number of *GhPYLs* and *GbPYLs* was roughly sum of *GaPYLs* and *GrPYLs*. These findings imply that each diploid species already contained about 20 PYL members prior to formulating *G. hirsutum* and *G. barbadense*, and the *GhPYLs* and *GbPYLs* mainly came from the allopolyploidization (*Wendel, Brubaker & Seelanan, 2010*).

The organizations of intron/exon and the number of exons in the surveyed *Gossypium PYLs* were very similar to those in the orthologs in rice, maize, tomato, rubber tree and *Brachypodium distachyon* (*González-Guzmán et al., 2014*; *He et al., 2014*; *Fan et al., 2016*; *Guo et al., 2017*; *Zhang et al., 2017*). Moreover, all of PYLs in *Gossypium* had similar helix-grip structures consisting of seven β-sheets, three α-helices and four loops, consistent with PYLs in *Arabidopsis*, rice and other plant species (*Ma et al., 2009*; *González-Guzmán et al., 2014*; *He et al., 2014*; *Fan et al., 2016*; *Guo et al., 2017*; *Zhang et al., 2017*). These results hint that these PYLs from different plant species have very similar functions and action modes.

Most *Gossypium PYLs* in subfamily I and II had no intron. However, some members in the two subfamilies such as *GrPYR1-3*, *GhPYL2-4A*, *GrPYL4-2* and *GbPYL4-1A* contained one or two introns (Fig. 2), suggesting that these genes obtain new introns during evolution.

Colinearity results revealed that approximate 400 homologous gene pairs existed among *PYLs* from *G. arboreum*, *G. raimondii* and *G. hirsutum* (Fig. 4), and numerous PYL gene pairs like *GbPYL2-2D* and *GbPYL2-2D'* had nearly identical amino acid sequences (Table S6). These results suggest that PYL family expands through segmental duplication events during evolution.

To uncover the homologous relationships of PYL gene family among different taxa, a phylogenetic tree was constructed based on the PYL protein sequences from four cotton species, *T. cacao*, *R. communis*, *V. vinifera*, *B. distachyon*, rice and *Arabidopsis* (Fig. 6). We observed *Gossypium* PYLs clustered more closely with cocoa *PYLs* than *OsPYLs*, in agreement with the evolutionary relationships among these plants.

Expression analysis results revealed that great majority of *GhPYLs* were expressed in various tissues like roots, stems, leaves, flowers and fibers. Notably, a large number of genes were highly expressed in the flower (Fig. 7). These results imply that ABA is implicated in modulation of reproductive development in cotton. Moreover, we noticed that some genes were preferentially expressed in roots, stems and fiber, pointing to the important roles of ABA in these tissues. PYLs have been documented to express in diverse tissues in various plants such as rice, maize and rubber tree (*Tian et al., 2015*; *Fan et al., 2016*; *Guo et al., 2017*). In rice, most *OsPYLs* were expressed in all tissues, *OsPYL3* and *OsPYL5* were predominantly expressed in leaves, and *OsPYL1* in roots (*Tian et al., 2015*). In maize, *PYL11* was upregulated in leaves, *PYL6* and *PYL10* in roots (*Fan et al., 2016*). In rubber tree, five genes were detected to express in all tissues tested, four genes were preferentially expressed in leaves, four in roots and one in flowers (*Guo et al., 2017*). These results were consistent with our findings in cotton, indicating the diverse biological functions of different PYLs in plants.

Changes in the expression levels of 40 *GhPYLs* were investigated in roots in responding to ABA or osmotic stress. The transcriptional abundances of many genes decreased after ABA treatments (Fig. 8), but increased post PEG treatments (Fig. 9), pointing to the different functional mechanisms of *GhPYLs* in response to ABA and osmotic stress. *Tian et al. (2015)* reported OsPYLs are differentially expressed after ABA treatment. Some AtPYLs have been addressed to be down-regulated by ABA treatment in *Arabidopsis* (*Santiago et al., 2012*). In *B. distachyon*, *BdPYL11* is down-regulated under ABA and PEG6000 treatments while *BdPP2CA4*, *BdPP2CA5*, *BdPP2CA6* and *BdPP2CA8* are up-regulated (*Zhang et al., 2017*). *ZmPYL4-11* was found to be down-regulated by ABA (*Fan et al., 2016*). *Bai et al.*

*(2013)* demonstrated that expression of many *PYLs* is reduced while the transcriptional abundances of some genes are enhanced upon ABA treatment in soybean. Together, these results indicate different *PYL* members have differential expression patterns in responding to ABA or osmotic stress, reflecting the diversity of *PYLs* in plants.

The interactions between 25 GhPYL members and GhABI1A or GhABI1D were analyzed by two-yeast hybrid method in the absence or presence of ABA. The results showed that GhPYLs selectively and specifically interacted with the two GhABI1s. Furthermore, these interactions were ABA-dependent or ABA-independent (Fig. 10). These results suggest that the GhPYLs should be functional ABA receptors in cotton. In *Arabidopsis*, AtPYL6, and AtPYL9 inhibit PP2Cs in a ABA-independent manner (*Hao et al., 2011*; *Miyakawa et al., 2013*). Consistently, a number of homologs of AtPYL6 and AtPYL9 such as GhPYL6-2D, GhPYL9-4D and Gh9-6A interacted with GhABI1A or GhABI1D without ABA, implying the conserved mechanism of ABA receptors between *Arabidopsis* and cotton. However, some cotton homologs of AtPYL4 (GhPYL4-2A) could interact with GhABIs in the absence of ABA whereas some other homologs of AtPYL4 (GhPYL4-1A, GhPYL4-2D) interplayed with GhABIs relying on ABA, although AtPYL4 suppresses PP2Cs without ABA (*Hao et al., 2011*; *Miyakawa et al., 2013*). These data suggest that the action modes of some ABA recepors of cotton may be altered during evolution. The detailed mechanisms need to be studied in the future.

## CONCLUSIONS

A total of 21, 20, 40 and 39 PYL genes were identified from *G. arboretum*, *G. raimondii*, *G. hirsutum* and *G. barbadense*, respectively. High commonality of gene structure, amino acid sequences and synteny of PYL members were found among the four surveyed *Gossypium* species. Specific expression patterns in tissues and diverse expression profiles in response to ABA and osmotic stress were uncovered. The interactions between 25 GhPYLs and two GhABI proteins were also investigated. These results suggest that the *Gossypium* PYLs are diverse functional ABA receptors. Our results gave a comprehensive information of the *Gossypium* PYLs for further research towards understanding the roles of PYLs in cotton.

## ACKNOWLEDGEMENTS

The authors thank Dr. Zhen Zhang in Institute of Cotton Research in Chinese Academy of Agricultural Sciences for providing good suggestions.

### Funding

This work was supported by the Science and Technology Development Program of He'nan in China (162102110005), Foundation of He'nan Educational Committee of China (15A210018, 17A180018 and 14B180029) and the ''111'' Project. The funders had no role in study design, data collection and analysis, decision to publish, or preparation of the manuscript.

## Grant Disclosures

The following grant information was disclosed by the authors:
Science and Technology Development Program of He'nan in China: 162102110005.
Foundation of He'nan Educational Committee of China: 15A210018, 17A180018, 14B180029.
"111" Project.

## Competing Interests

The authors declare there are no competing interests.

## Author Contributions

- Gaofeng Zhang and Tingting Lu conceived and designed the experiments, performed the experiments, analyzed the data, wrote the paper, prepared figures and/or tables, reviewed drafts of the paper.
- Wenwen Miao and Mi Tian contributed reagents/materials/analysis tools.
- Lirong Sun analyzed the data, contributed reagents/materials/analysis tools.
- Ji Wang performed the experiments, contributed reagents/materials/analysis tools.
- Fushun Hao conceived and designed the experiments, contributed reagents/materials/analysis tools, wrote the paper, prepared figures and/or tables, reviewed drafts of the paper.

## Data Availability

   The raw data has been provided as a Supplemental File.

## Supplemental Information

Supplemental information for this article can be found online at http://dx.doi.org/10.7717/peerj.4126#supplemental-information.

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
