# Peer review of "Genome-wide identification of ABA receptor PYL family and expression analysis of PYLs in response to ABA and osmotic stress in Gossypium"

_PeerJ, doi:10.7717/peerj.4126_

## Round 0.1 · original submission · Minor Revisions

· Academic Editor

Minor Revisions

I agree with the comments and suggestions from the reviewers. Please make sure they are all incorporated in the revised manuscript. In addition to the reviewers' comments, I would like you to do the following as a must to get the manuscript accepted.

* for the first part of the introduction where you describe the overall schema of interactions, add a new figure explaining these interactions for our readers to understand various events and processes being discussed.

* Your table-1 can be a supplelemntary table.

*Also suggested by another reviewer, add a new gene nomenclature and homolog table that lists all the genes PYL genes identified in all the four cotton and additional 6 species. The table must contain the respective gene names/symbols, gene IDs with homologs side by side in the same row showing the best matches from each species including the percent identity. Add columns for each species.

* provide the sequence fasta format file with all the PYL genes

* For the figures 1 and 5, submit the sequence alignment files and the actual phylogenetic tree (nwk format) that also has branch lengths and bootstrap values as supplementary tables.

*submit the PYL primer sequences to NCBI's probeDB and also mention in the method. Feel free to add primer sequences and the respective gene IDs and symbols in a new table.

Reviewer 1 ·

Basic reporting

No comment.

Experimental design

No comment.

Validity of the findings

No comment.

Additional comments

The manuscript submit by Zhang et al. regarding genome wide identification of ABA receptor protein family PYLs in four different species of cotton is an extensive report. Though the PYL family members are reported from many other plants, but the current report is the first from Gossypium spp. In spite of no novelty in the entire research the information generated will be of interest for researchers working with Gossypium spp. Apart from identifying the entire family members from progenitor G. arboretum, G. raimondii they also identified them in the tetraploid G. hirsutum and G. barbadense species. The authors also performed the chromosomal location of each gene, synteny among all four species and phylogeny with PYLs of other known plants. The members of G. hirsutum was further investigated for its tissue specific expression, regulation under ABA treatment and osmotic stress. They also analysed GhPYLs interaction with two members PP2C proteins, GhABI1A and GsABI1D known to be involved in ABA signalling. Even though, the later part of the work is fragmented and ends abruptly, the information in the manuscript is interesting.
I have following minor concerns for the submitted work:
The nomenclature of the Gossypium PYLs are solely based on that of Arabidopsis PYLs. It seems this nomenclature (giving a specific number at the end) does not hold true when PYLs of rice is also considered. Can author take this opportunity to propose a uniform nomenclature of PYLs across different plant species.
Tissue specific expression of GhPYLs are compared with the expression of PYLs in root (Fig. 6). As such the modulation of these genes could not be assessed in root. It will be better if an house keeping gene is taken as internal control.
The Y2H analysis result may also be presented in tabular form, that will be easy to see in one glance which PYLs are interacting with the two selected proteins.

Reviewer 2 ·

Basic reporting

The authors have identified the PYL genes in Gossypium species. They have identified 21 genes in G.arboreum, 20 in G.raimondii, 40 in G.hirsutum and 39 in G barbadense. They have done thorough bioinformatics analysis of these genes, and have identified conserved secondary structure that are essential for the function for ABA receptors in arab.
They have analysed the G.hirsutum genes for the relative gene expression in osmotic stress and in response to ABA. Using two hybrid system they have also shown the interaction of GhPYLs and GhABI1A and GhABI1D.

Experimental design

The material and methods are clearly written.

Validity of the findings

The authors have done several experiments to validate the claims made in the manuscript. The relative gene expression shows the role of PYL in ABA response and osmotic stress. Similarly the yeast two hybrid analysis shows the interaction of PYL with Gh ABi1A and Gh ABI1D.

Additional comments

There was one paper by Chen et al published in PPB on the Overexpression of cotton PYL genes in Arabidopsis enhances the transgenic plant tolerance to drought stress. The authors have not mentioned this in their manuscript and since this paper is on PYL in G. hirsutum, the authors need to revise the manuscript accordingly.

Reviewer 3 ·

Basic reporting

No comment

Experimental design

Details were missed in some experiments. For example:

Line 148:Which parts of the plants were sprayed with ABA? How to treat with 10% PEG6000?
The paragraph of Line 144: You have both treated plants (after Line 145 which is clear to readers) and untreated plants (before line 145). Please make it clearer that the plants were not treated. The roots of the treated plants were sampled and frozen. How were the samples of the untreated plants treated after sampled?
Line 150: How many grams of the samples were used for RNA extraction? How were the samples homogenized before the extraction?
Line 151 and line 152: The reaction conditions of cDNA synthesis of RT-PCR? Please rewrite this part.
Line 155: “Experiments were repeated at least three biological replicates.” Do you mean the experiments were repeated at least three times? How many replicates did you have each time?
Line 154 primer: please provide primer sequences in Materials and Methods or in the supplementary

Validity of the findings

No comment

Additional comments

Generally, the manuscript is written in professional English with minor errors. The study is novel and meaningful. The introduction has sufficient background information.The hypotheses and experiment design are appropriate. But, details were missed in some experiments. The result section including figures and tables are generally OK. But, 1) the explanation of some results are not clear or sufficient; 2) the figure may not be a very good format to present the results shown in Figure 6, 7, and 8. The discussion is OK.
Please read the detailed comments and suggestions in the annotated PDF manuscript.

Annotated reviews are not available for download in order to protect the identity of reviewers who chose to remain anonymous.

---

## Round 0.2 · Minor Revisions

· Academic Editor

Minor Revisions

Please bring the Figure S1 into the main manuscript. This is an opportunity for you to review the literature showing the advances in the PYL story. See the figure-1 in https://www.frontiersin.org/articles/10.3389/fpls.2015.00088/full is appreciated. Do not copy, but add new information and improvise. Your Figure S1 is very confusing and does not convey the things appropriately in the introduction text.

Submit fasta sequence, alignment and tree flies as supplementary files and not marked as supplementary tables.

Provide the accesion# of the primer sequences submitted to NCBI

In table-1 in addition to gene symbol also provide the gene ID

Show the data from UBQ7 controls to support your expression profiles for comparison.

In addition to the gene symbol Include gene IDs in the tree shown in Figure-2 and the Newick tree file. Same goes to the sequence file and the fasta alignment files.

Bring Table S8 homolog tale into the main manuscript. That way readers will able to comprehend the data more appropriately. However, your current Table S8 is a total mess. Reformat the table in the following column order to show your gene sets and nomenclature.

name cotton species-1
cotton species-1 gene ID (gene symbol)--symbol goes in parenthesis for this and following columns
Cotton species-1 homeolog/paralog gene ID (gene symbol). Separate multiple by pipe or comma
cotton species-2 homolog gene ID & symbol (including homeologs and paralogs; separate multiple by pipe or comma)
cotton species-3 homolog gene ID & symbol (including homeologs and paralogs; separate multiple by pipe or comma)
cotton species-4 homolog gene ID & symbol (including homeologs and paralogs; separate multiple by pipe or comma)
Arabidopsis homolog gene id and (gene symbol)
Rice homolog gene id and (gene symbol)
and repeat for any other species the given gene symbol and the gene id.

---

## Round 0.3 · Minor Revisions

· Academic Editor

Minor Revisions

Since new figures and tables were added to the main manuscript, please make sure you have corrected the figure numbers and tables in the main manuscript. Same goes for supplementary files. Please check and confirm throughout.

---

## Round 0.4 · accepted · Accept

· Academic Editor

Accept

Many thanks for making all the necessary changes.